# Assessment of non-alcoholic fatty liver disease (NAFLD) severity with novel serum-based markers: A pilot study

Atul Goyale[1], Anjly Jain[1], Colette Smith[2], Margarita Papatheodoridi [3], Marta Guerrero Misas[3], Davide Roccarina[3], Laura Iogna Prat[3], Dimitri P. Mikhailidis[1], Devaki Nair[1☯], Emmanuel Tsochatzis[3☯]*

1 Department of Clinical Biochemistry, Royal Free Hospital, London, United Kingdom, 2 UCL Research Department of Infection & Population Health, Royal Free Hospital, London, United Kingdom, 3 UCL Institute for Liver and Digestive Health, Royal Free Hospital and UCL, London, United Kingdom

☯ These authors contributed equally to this work.
* e.tsochatzis@ucl.ac.uk

**Data Availability Statement:** All relevant data are in the manuscript and its Supporting information files.

## Abstract

### Background/Aims

Non-alcoholic fatty liver disease (NAFLD) represents a significant public health issue. Identifying patients with simple steatosis from those with non-alcoholic steatohepatitis (NASH) is crucial since NASH is correlated with increased morbidity and mortality. Serum-based markers, including adipokines and cytokines, are important in the pathogenesis and progression of NAFLD. Here we assessed the usefulness of such markers in patients with NAFLD.

### Methods

This prospective, cross-sectional study included 105 adult patients with varying severity of NAFLD. Twelve serum-based markers were measured by 3 biochip platforms and 2 enzyme-linked immunosorbent assay (ELISA) methods. We also developed a NAFLD individual fibrosis index (NIFI) using the serum-based markers mostly correlated with fibrosis severity.

### Results

Sixty-one out of 105 patients were male (58.1%) with mean age was 53.5 years. Higher Interleukin-6 (IL-6) increased (p = 0.0321) and lower Matrix Metalloproteinase-9 (MMP-9) serum levels (p = 0.0031) were associated with higher fibrosis as measured by Fibroscan® in multivariable regression analysis. Using receiver-operating characteristic (ROC) curve analysis for the NIFI, area under the curve for predicting Fibroscan values ≥ 7.2 kPa was 0.77 (95%CI: 0.67, 0.88, p<0.001), with sensitivity of 89.3%, specificity of 57.9% and a positive likelihood ratio of 2.8.

**Funding:** The authors received no specific funding for this work.

**Competing interests:** The authors have declared that no competing interests exist.

## Conclusions

Increasing fibrosis severity in NAFLD is associated with differential expression of IL-6 and MMP-9. NIFI could be valuable for the prediction of advanced NAFLD fibrosis and potentially help avoid unnecessary interventions such as liver biopsy in low-risk patients.

## Introduction

Non-alcoholic fatty liver disease (NAFLD) is an increasing health issue with a world-wide prevalence of around 25% [1]. In NAFLD, there is accumulation of fat in the liver (>5%) in the absence of other liver disease (e.g. viral and auto-immune hepatitis), significant alcohol consumption, use of steatogenic medication (e.g. amiodarone, methotrexate or isoniazid), and hereditary disorders [2].

### NAFLD pathogenesis

The "multiple hit" hypothesis of NAFLD pathogenesis [3] suggests that energy imbalances attributable to excess calorie intake and insufficient exercise lead to increased insulin resistance which serves as a 'key hit' on genetically predisposed subjects to induce NAFLD. Further 'hits' consist of a combination of hormones secreted from adipose tissue, nutritional factors, gut microbiota in-addition to genetic and epigenetic factors that then contributed to oxidative stress and inflammation aiding the progression of steatosis to non-alcoholic steatohepatitis (NASH) [4]. An estimated 5% of the United Kingdom (UK) population are affected by NASH [5]. Persistent inflammation results in scar tissue formation and progression to fibrosis, which then may progress further to liver cirrhosis, with end-stage liver disease and hepatocellular carcinoma being possible outcomes [6].

Most patients may not progress beyond simple steatosis when picked up early and managed appropriately [5]. Differentiating simple steatosis from NASH is essential in the clinical setting, as simple steatosis has a benign course whereas NASH, in sharp distinction, is associated with reduced life expectancy [7]. Besides, there is increased cardiovascular (CV) risk [8] with myocardial infarction and cerebral vascular accident being the leading causes of morbidity and mortality in the NAFLD population [1].

### NAFLD diagnosis and assessment of disease severity

Several diagnostic panels based on a combination of liver tests and clinical parameters such as the NAFLD fibrosis score (age, body mass index (BMI), diabetes status, with aspartate aminotransferase (AST)/alanine aminotransferase (ALT) ratio, platelet count and albumin) and FIB-4 score (age, BMI, AST/ALT ratio and platelet count) have been developed to facilitate the non-invasive assessment of NAFLD severity [9]. The Enhanced Liver Fibrosis (ELF) test is an algorithm that combines quantitative serum concentration measurements of three extracellular matrix fibrosis markers (tissue inhibitor of metalloproteinases-1, amino-terminal propeptide of type III procollagen and hyaluronic acid) [10] that has shown good correlation with fibrosis stages in chronic liver disease in general and NAFLD specifically [11]. FIB-4 score and the ELF test are used as decision tools in primary care to determine which patients with suspected NAFLD are referred to secondary care, with the latter being utilised when result of the former is equivocal [12].

Although these scoring systems have a greater utility in the detection of advanced fibrosis than liver tests alone, in practice they rarely negate the need for a liver biopsy, which despite its invasive nature and limitations, remains the gold standard [13]. Liver biopsy limitations

include the cost as well as the expertise required in obtaining and interpreting a liver biopsy. Additionally, as fibrosis is not evenly distributed throughout the liver, biopsy sampling issues can potentially give rise to incorrect disease severity categorisation. Other considerations include patient acceptance of an invasive procedure and potential complications such pain and bleeding [13].

FibroScan® is an accurate and accepted non-invasive tool that employs sound wave technology to measure liver tissue elasticity. It is used in secondary care to evaluate liver steatosis and fibrosis in patients with NAFLD and has shown concordance with liver biopsy results [14]. In practice, NAFLD disease severity is assessed by a combination of the non-invasive clinical, biochemical and sonographic parameters mentioned above with liver biopsy being reserved for patients with suspected progressive or advanced disease.

## Role of adipokines and cytokines in NAFLD

The 'multi-hit' hypothesis of NAFLD pathogenesis raises the possibility that several potentially quantifiable biochemical imbalances may be present and could be useful clinically. Adipose tissue, now accepted as a metabolically active endocrine organ [15], releases a variety of bioactive cytokines, termed adipokines into the blood stream [7]. These adipokines, central to the communication between adipose tissue and other organs, play a role in the initiation of NAFLD. Adipokines produced by adipose tissue including adiponectin, leptin and resistin play an important role in energy homeostasis [15].

Pro-inflammatory cytokines such as tumour necrosis factor alpha (TNFα) and interleukin-6 (IL-6) are produced by adipocytes, hepatic stellate cells and Kupffer cells as a result of hepatic fat accumulation [16]. These cytokines have immune-modelling functions utilizing signalling pathways involving nuclear receptors such as peroxisome proliferators-activated receptors (PPARs) and play a role in the evolution of NAFLD [16].

Several further related markers [soluble TNF-alpha receptor-1 (sTNFR1), soluble TNF-alpha receptor-2 (sTNFR2), soluble IL-6 receptor (sIL-6R), ghrelin, plasminogen activator inhibitor-1 (PAI-1), cytokeratin-18 fragments, and matrix metalloproteinase-9 (MMP-9)] could play a significant role in NAFLD pathogenesis. These metabolic markers have been the subject of research, often with conflicting results [9,17–28]. These markers were evaluated in the present study.

The aim of the present study was to assess the serum levels of 12 metabolic markers [adiponectin, leptin, ghrelin, TNFα, IL-6, PAI-1, cytokeratin-18 fragments, resistin, sIL-6R, sTNFR1, sTNFR2 and MMP-9] in patients with NAFLD and to investigate the potential association of these markers with the severity of fibrosis.

A secondary aim was to generate a predictive NAFLD individual fibrosis index (NIFI) based on the markers that showed strongest correlation (positive or negative) with NAFLD severity. Such an index could be used as a predicator of disease severity and possibly help negate the need for liver biopsy in patients with low scores. The NIFI may also help predict progression of NAFLD and any effect of treatment.

## Materials and methods

The present prospective, cross-sectional study was performed at The Royal Free London NHS Foundation Trust, a large teaching hospital with a specialist hepatology service. The study, part of a wider study of non-invasive assessment and determinants of liver fibrosis in NAFLD was approved by the University College Hospital: Royal Free Hospital Ethics committee (Reference number: NC2014.006, project as part of the UCL-RFH Biobank Research Tissue Bank, REC reference: 16/WA/0289, Wales Research Ethics Committee 4). Written consent was obtained

from all participants. All patients were provided with written information regarding the aims and objective of the study. Blood samples for the measurement of the 12 serum-based markers were collected on enrolment, as part of the study protocol.

## Population

Patients (n = 105) with NAFLD attending the Specialist Multidisciplinary NAFLD Liver clinic were enrolled. The diagnosis of NAFLD was based on the presence of fatty liver on ultrasound in patients, who had no history of alcohol misuse (defined as >14 units/week in females or >21 units/week in males) and in whom other causes of liver disease (viral, auto-immune, hereditary haemochromatosis, Wilson's disease and alcoholic hepatitis) had been excluded with a comprehensive work up. Patients with liver conditions other than NALFD were not included in the present study. Patients with concomitant systemic inflammatory conditions, including rheumatoid arthritis, psoriasis and inflammatory bowel disease were excluded from the study as were patients on immunosuppressive therapy. All included patients were in a stable clinical condition.

Data collected included height, weight, renal, liver, bone and lipid profiles as well as a full blood count. Demographic data and co-morbidities (history of conditions such as diabetes mellitus, hypertension, hyperlipidaemia or a history of CV disease) were recorded for all patients. Patients also underwent a liver ultrasound and a fibroscan. Severity of NAFLD was characterized by a combination of clinical, laboratory, radiological and fibroscan findings. Where indicated, patients proceeded to liver biopsy for a histological diagnosis.

## Anthropometry assessment

BMI was calculated as weight (in kg)/height (in $m^2$) [29].

## General investigations

Liver enzymes and lipids were analysed using standard methods on a Roche Modular P$^®$ analyser and Roche$^®$ reagents (Roche Diagnostics, GmbH, D-68298 Mannheim, Germany). Platelets were analysed using standard methods on one of 4 Sysmex XN-9000$^™$ analysers (Sysmex America, Illinois, U.S.A). FIB-4 was calculated using the formula [9].

$$FIB - 4 = (Age[years] x\ AST[U/L])/(Platelet\ count[10^9/L]) x (\sqrt{ALT[U/L]})$$

## Fibroscan$^®$

Fibroscan$^®$ (Echosens, Paris, France) was carried out by experienced examiners (DR and LIP) following a standard protocol in all patients. The median liver stiffness of 10 successful measurements was noted in (kPa). Fibroscan measurements of ≥7.2 kPa were used do define the likely presence of significant fibrosis [30]. The M or XL probe were used as recommended by the device, according to the patient's anthropometric characteristics, in order to obtain the most optimal measurements. Moreover, controlled attenuated parameter (CAP) has been measured as a marker of steatosis and the cut-off above 302 dB/m has been used to predict steatosis >S1 [31].

## Measurement of serum concentrations of serum-based markers

In the present study, blood samples for metabolic marker measurements were collected from consented patients in addition to the blood tests taken as part of their clinic review. Patients were provided information pertaining to the study verbally and in writing, through a detailed patient information leaflet. All blood samples were measured in the morning following an

overnight fast. Plasma and serum were separated by centrifugation for 10 min at 3000 revolutions/min (rpm) and then stored at -70˚C until the time of analysis. All samples were given a unique non-patient identifiable barcode, ensuring blinding of the researcher.

## Biochip assay

Three separate biochip technology systems employing the Randox Investigator™ were used in this study to measure 10 of the metabolic markers in patients' stored serum samples. The Metabolic Syndrome Array I (leptin, resistin, TNFα, IL-6 and PAI-1), Metabolic Syndrome Array II (adiponectin) and Cytokine Array IV (sIL-6R, sTNFR1, sTNFR2 and MMP-9).

## Ghrelin assay

Ghrelin concentrations were measured by competitive solid-phase sandwich enzyme-linked immunosorbent assay (ELISA) tests as per the instructions of the manufacturer (RayBiotech, Norcross, USA) in patients' stored serum samples.

## Keratin-18 assay

Keratin-18 concentrations were measured by quantitative solid-phase sandwich enzyme immunoassay assay (ELISA) tests for the M30 antibody, according to the instructions of the manufacturer (PEVIVA, AB, Sweden) in patients' stored serum samples. We preferred the M30 to the M65 antibody as the former is specific for apoptotic fragments of keratin-18.

## Power calculation

A power calculation for sample size (n) using the formula $n = 2\sigma^2 (z_{1-\alpha/2} + z_{1-\beta})^2/(\mu_1 - \mu_2)^2$ was performed. Based on confidence level of 95% corresponding to a $z_{1-\alpha/2}$ value of 2.87 for a p of 0.004. The $z_{1-\alpha/2}$ value and the p value we selected were adjusted for 12 separate serum-based markers. A sample size calculation with a power of 80%, giving a value for $z_{1-\beta}$ of 0.84 revealed that an initial sample of 90 patients could be sufficient to achieve statistically significant when comparing the relationship between biomarker concentrations and NAFLD severity. The sample size calculation was based work by Kumar et al. who found higher IL-6 concentrations in patients with more advanced stages of fibrosis; stage III fibrosis compared with stage I fibrosis (560 [523–575] compared with 206 [181–413] pg/ml, respectively [32].

## Statistical analysis

Analyses were performed using SAS Version 9.3 (SAS Institute Inc, Cary, NC). The data was checked for normality by Shapiro Wilk test in addition to visual inspection of the distribution of the data. Where required the data was logged and further analysis was performed. Normally distributed numerical data are expressed as mean (standard deviation). For describing the levels of the serum markers across different groups, geometric means (95% confidence intervals, CI) were used. Geometric means are calculated by raising the product of a series of values of the variable to the inverse of the total length of the series. Data that where not normally distributed are reported as median and interquartile range (IQR). Categorical data are presented as number and percentages of patients. Pearson's chi square test was used for categorical data. Spearman rank test was used for correlations; a two-sided p-value of <0.05 was considered significant. A receiver-operating characteristic (ROC) curve was generated for NIFI against Fibroscan measurements of ≥7.2 kPa in order to generate optimal sensitivity and specificity cut-offs for NIFI.

## Results

### Demographics and clinical characteristics

Patients (n = 105) with a clinical diagnosis of NAFLD were included in this study. Baseline demographic data and clinical characteristics are shown in Table 1. The mean age was 53.5 ± 13.4 years; 61 patients were male (58.1%). The mean BMI was 32.4 ± 6.1 kg/m$^2$. The rates of pre-existing co-morbidities were as follows: 31 (29.5%) patients had diabetes mellitus, 52 (49.5%) patients had hyperlipidaemia (defined as total cholesterol > 5mmol/L or low-density lipoprotein (LDL) > 3mmol/L) and 50 (47.6%) had hypertension. There was a history of CV disease in 10 (9.5%) patients.

### Metabolic markers and demographic characteristics

Adiponectin geometric mean was higher in those aged ≥50 and females. Leptin concentrations were higher in females. IL-6 was higher in persons aged ≥ 50. Resistin, TNFα, PAI-1, sIL-6R, sTNFR1, sTNFR2, MMP9, keratin-18 and ghrelin concentrations did not display any differences with age, BMI or gender (S1 Table).

### Correlations of serum-based markers with ALT, AST and Fib-4 score

Although ALT and AST abnormalities are not associated with the presence and severity of NAFLD, a quantitative correlation might still be of value as it can expose trends and associations for further investigation.

Adiponectin and leptin did not significantly correlate with ALT, AST or FIB-4 score. Concentrations of resistin, TNFα, IL-6 and ghrelin positively correlated with FIB-4 scores (r = 0.240 [p = 0.030], 0.225 [p = 0.042], 0.276 [p = 0.012] and 0.285 [p = 0.0075]), respectively. However, there was no correlation between these 4 metabolic markers with the ALT or AST activities, from which FIB-4 is generated. The relationship between the 12 metabolic markers and ALT, AST and FIB-4 is displayed in Table 2. Significant negative correlation was found between sIL-6R and AST concentrations (r = -0.243, [p = 0.029]). TNF-alpha receptors 1 and 2

**Table 1. Demographic and clinical characteristics of all study patients.**

| Characteristics | All patients, N = 105 |
|---|---|
| Male gender, n (%) | 61 (58.1%) |
| Age, years | 53.5 ±13.4 |
| Age <50 years, n (%) | 43 (41.0%) |
| BMI, kg/m$^2$ | 32.4 ± 6.1 |
| BMI <30 kg/m$^2$, n (%) | 45 (42.9%) |
| Diabetes mellitus type II, n (%) | 31 (29.5%) |
| Hyperlipidaemia, n (%) | 52 (49.5%) |
| Hypertension, n (%) | 50 (47.6%) |
| History of CVD, n (%) | 10 (9.5%) |
| AST:ALT ratio > 1.00, n (%) | 16 (15.2%) |
| NAFLD Fibrosis Score > -1.455, n (%) | 45 (42.9%) |
| FIB-4 score >1.3, n (%) | 42 (47.7%) |
| Liver stiffness by Fibroscan ≥7.2 kPa, n (%) | 47 (44.8%) |
| CAP measurements >302 dB/m, n (%) | 38 (71.7%) |

Quantitative variables are presented as Mean ± Standard deviation, and categorical variables as count, n (%).
BMI, Body Mass Index; CVD, cardiovascular disease; AST, aspartate aminotransferase; ALT, alanine aminotransferase; NAFLD, non-alcoholic liver disease; kPa, kilopascal.

showed significant negative correlation ALT (r = -0.337 [p = 0.0019] and r = -0.348 [p = 0.0013, respectively]). MMP-9 showed significant negative correlations with ALT, AST and FIB-4 (r = -0.288 [p = 0.0087], -0354 [p = 0.001] and -0.327 [p = 0.0027], respectively). PAI-1 showed positive correlation with ALT (r = 0.224 [p = 0.0433]) and negative correlation with Fib-4 (r = -0.326 [p = 0.0028]). Keratin-18 displayed strong positive correlation with ALT (r = 0.462 [p < 0.0001]) and AST (r = 0.448 [p < 0.0001]).

## Serum-based markers and Fibroscan result

Fibroscan results ≥7.2 kPa were significantly associated with increased geometric means of ghrelin, TNFα and IL-6 (p = 0.018, 0.022 and 0.008, respectively). The relationship between the 12 serum-based markers and Fibroscan results is shown in (S2a Table). Lower concentrations of MMP9 were significantly associated with elevated Fibroscan readings (p = 0.005). No significant association was found between Fibroscan results and concentrations of adiponectin, leptin, keratin-18, resistin or any of the 3 soluble receptors (sIL-6R, sTNFR1, and sTNFR2).

Moreover, the levels of the 12 serum-based markers were examined in relation to the CAP values; while MMP-9 levels were significantly lower, keratin-18 and ghrelin levels were significantly higher in patients with steatosis >S1 (CAP ≥302dB/m) than those without (S2b Table).

## Fibroscan result and demographic characteristics

Multi-variable analysis revealed that age ≥50 years and diabetes were associated with Fibroscan results of ≥7.2 kPa (p = 0.013 and 0.042, respectively). Gender, BMI and history of hyperlipidaemia, hypertension and CV disease did not meet significance in their association with Fibroscan in multivariate analysis, see Table 3a.

**Table 2. Correlations between aspartate aminotransferase (AST), alanine aminotransferase (ALT) serum levels and FIB-4 score with various metabolic markers in 105 study patients.** Spearman correlation coefficients (p-value) are presented for each marker and significant p- values are shown in **BOLD font.**

|  | ALT (U/L) | AST (U/L) | FIB-4 score |
|---|---|---|---|
| **Median (IQR)** | 48 (30, 74) | 33 (28, 46) | 1.19 (0.77, 1.73) |
| **Correlation coefficients with metabolic markers** | | | |
| **Adiponectin (µg/ml)** | -0.187 (0.096) | -0.038 (0.740) | 0.108 (0.340) |
| **Leptin (ng/ml)** | -0.015 (0.900) | 0.071 (0.530) | 0.034 (0.770) |
| **Resistin (ng/ml)** | -0.125 (0.260) | 0.054 (0.630) | 0.240 (**0.030**) |
| **TNFα (pg/ml)** | 0.090 (0.400) | 0.215 (0.054) | 0.225 (**0.042**) |
| **IL-6 (pg/ml)** | -0.151 (0.180) | 0.049 (0.660) | 0.276 (**0.012**) |
| **PAI-1 (ng/ml)** | 0.224 (**0.043**) | 0.088 (0.434) | -0.326 (**0.003**) |
| **sIL-6R (ng/ml)** | -0.197 (0.077) | -0.243 (**0.029**) | -0.174 (0.120) |
| **sTNFR1 (ng/ml)** | -0.337 (**0.002**) | -0.213 (0.056) | 0.020 (0.860) |
| **sTNFR2 (ng/ml)** | -0.348 (**0.001**) | -0.128 (0.260) | 0.180 (0.110) |
| **MMP-9 (ng/ml)** | -0.288 (**0.009**) | -0.354 (**0.001**) | -0.327 (**0.003**) |
| **Keratin-18 (U/L)** | 0.462 (**<0.001**) | 0.448 (**<0.001**) | 0.100 (0.370) |
| **Ghrelin (ng/ml)** | 0.052 (0.630) | 0.137 (0.210) | 0.285 (**0.008**) |

TNFα, Tumour Necrosis Factor alpha; IL-6, Interleukin-6; PAI-1, Plasminogen Activator Inhibitor-1; sIL-6R, Interleukin-6 receptor; sTNFR1, soluble TNFα receptor 1; sTNFR2, soluble TNFα receptor 2; MMP-9, Matrix Metalloproteinase-9.

**Table 3.** **a.** Factors associated with Fibroscan result ≥7.2 kPa. Logistic regression analysis in all study patients (n = 105). **b.** Multi-variable analysis of factors associated with Fibroscan result ≥7.2 kPa.

| Factors | | Univariable analysis | | Multivariable analysis | | Univariable | | | Multivariable | | |
|---|---|---|---|---|---|---|---|---|---|---|---|
| | | OR (95% CI) | p | OR (95% CI) | p | OR | 95% CI | p | OR | 95% CI | p |
| Sex | Male vs Female | 0.67 (0.30, 1.49) | 0.323 | 1.34 (0.47, 3.80) | 0.579 | 0.67 | 0.30, 1.49 | 0.323 | | | |
| Age (years) | <50 vs ≥50 | 0.20 (0.08, 0.50) | <0.001 | **0.24 (0.08, 0.74)** | **0.013** | **0.20** | **0.08, 0.50** | **0.0006** | **0.26** | **0.07, 0.74** | **0.014** |
| BMI (kg/m$^2$) | <30 vs ≥30 | 0.81 (0.23, 3.21) | 0.832 | 0.79 (0.18, 3.69) | 0.713 | 1.07 | 0.48, 2.40 | 0.870 | | | |
| Diabetes | Yes vs No | 4.40 (1.79, 10.83) | 0.001 | **2.98 (0.95, 9.38)** | **0.022** | **4.40** | **1.79, 10.83** | **0.001** | **4.57** | **1.20, 17.39** | **0.026** |
| Hyperlipidaemia | Yes vs No | 2.07 (0.92, 4.70) | 0.080 | 1.09 (0.38, 3.10) | 0.874 | **2.07** | **0.92, 4.70** | **0.080** | 0.66 | 0.20, 2.22 | 0.502 |
| Hypertension | Yes vs No | 2.71 (1.19, 6.20) | 0.018 | 1.29 (0.44, 3.79) | 0.638 | **2.71** | **1.19, 6.20** | **0.018** | 1.12 | 0.32, 3.93 | 0.857 |
| History of CVD | Yes vs No | 1.66 (0.39, 7.05) | 0.494 | 0.45 (0.08, 2.65) | 0.380 | 1.66 | 0.39, 7.05 | 0.494 | | | |
| Ghrelin | Per 1-log higher | | | | | **1.63** | **1.07, 2.50** | **0.024** | 1.13 | 0.63, 2.02 | 0.682 |
| TNFα | Per 1-log higher | | | | | **4.50** | **1.27, 15.93** | **0.019** | 2.20 | 0.49, 9.86 | 0.302 |
| IL-6 | Per 1-log higher | | | | | **2.34** | **1.25, 4.36** | **0.008** | **2.13** | **1.07, 4.25** | **0.032** |
| MMP-9 | Per 1-log higher | | | | | **0.34** | **0.15, 0.76** | **0.008** | **0.19** | **0.06, 0.57** | **0.003** |

OR, odds ratio; CI, confidence interval; BMI, Body Mass Index; CVD, cardiovascular disease.

kg, kilograms; m, metre; CVD, cardiovascular disease; TNFα, Tumour Necrosis Factor alpha; IL-6, Interleukin-6; MMP-9, Matrix Metalloproteinase-9.

## Results from logistic regression model: Multi-variable analysis of Fibroscan, metabolic markers and demographic characteristics

Multi-variable analysis of the relationship between Fibroscan results of ≥7.2 kPa, demographic data and metabolic markers is shown in Table 3b. In order to optimise the multivariate model, only variables with a p <0.10 in univariate analysis were processed in the multivariate model.

Age and history of diabetes were associated with Fibroscan results of ≥7.2 kPa (see Table 3a) in the multivariate model that only included demographic characteristics. Age, diabetes, IL-6 and MMP-9 were significantly associated with a stiffness of ≥7.2 kPa in the multivariate model that included both demographic factors and serum-based markers (ghrelin, IL-6, TNFα and MMP-9) (Table 3b).

We assessed the association of these 4 metabolic markers (ghrelin, IL-6, TNFα and MMP-9) and their direct association with demographic factors in the multivariable model (S3a–S3d Table) the only significant association found was with IL-6 concentration and increasing age (p = 0.009).

The results show that following multivariate analysis, Fibroscan results of ≥7.2 kPa remained significantly associated with increased IL-6 concentrations (p = 0.032) and lower concentrations of MMP-9 (p = 0.003).

We also examined whether these factors were associated with Fibroscan results ≥9.6 kPa, indicating advanced fibrosis (S4a and S4b Table). From the multivariable analysis examining only demographic characteristics associated with Fibroscan values ≥9.6 kPa, only age was significantly associated with advanced fibrosis (p = 0.026) (S4a Table), while when serum markers were also added to the model (S4b Table), both age and TNFa were significantly associated with Fibroscan results ≥9.6 kPa (p = 0.026 and p = 0.028, respectively) (S4b Table).

## Derivation of NIFI score

IL-6 and MMP-9 were combined to generate the NIFI. This index was derived in a similar way to the homeostatic model assessment for insulin resistance (HOMA-IR), which is based on

multiplying the glucose concentration by the insulin concentration and then dividing by a factor (product of mean glucose and insulin values), which normalises to 1.0 [33]. In the current study, as Fibroscan results of ≥7.2 kPa were associated with increased IL-6 concentrations and decreased MMP-9 concentrations, NIFI was calculated by deriving the mean of the reference ranges of IL-6 and MMP-9 [34,35]. The quotient of these two numbers 0.026 (rounded to 2 significant figures), which formed the basis for the divisor to derive a 'NIFI', defined as the quotient of IL-6 and MMP-9 divided by 0.026.

Using ROC analysis for NIFI, the cuff-off that gave optimal sensitivity and specificity for significant fibrosis was 1.45, with a likelihood ratio of 2.8. At this cut off, sensitivity and specificity were 89.3% and 57.9%, respectively (Fig 1). With ROC curve analysis we can establish a rule-in 90% specificity NIFI cut-off of 3.79 and a rule-out 90% sensitivity NIFI cut-off of 1.41. Of the total 105 patients in the current study, the NIFI of 26 patients (24.8%) was between these two cut-offs, while the proportion of patients with indeterminate results for FIB-4 and NFS were 35 (32.7%) and 24 (22.4%), respectively.

Finally, we examined the diagnostic accuracy of the NIFI cut-offs for significant fibrosis, in comparison to the recommended FIB-4 and NFS dual cut-offs (<1.3/ >2.67 and <-1.455/ >0.675, respectively). Of note, FIB-4 cut-offs <1.3 and >2.67 had 66% rule-out sensitivity and 100% rule-in specificity, respectively, as did the NFS dual cut-offs <1.455 and >0.675 for the prediction of significant fibrosis ≥ 7.2kPa.

## Discussion

### Metabolic markers and NAFLD severity as defined by Fibroscan ≥7.2 kPa

This study investigates a panel of 12 markers in adult NAFLD patients using 3 separate multianalyte platforms as well as 2 ELISA kits. Differences were found in concentrations for a

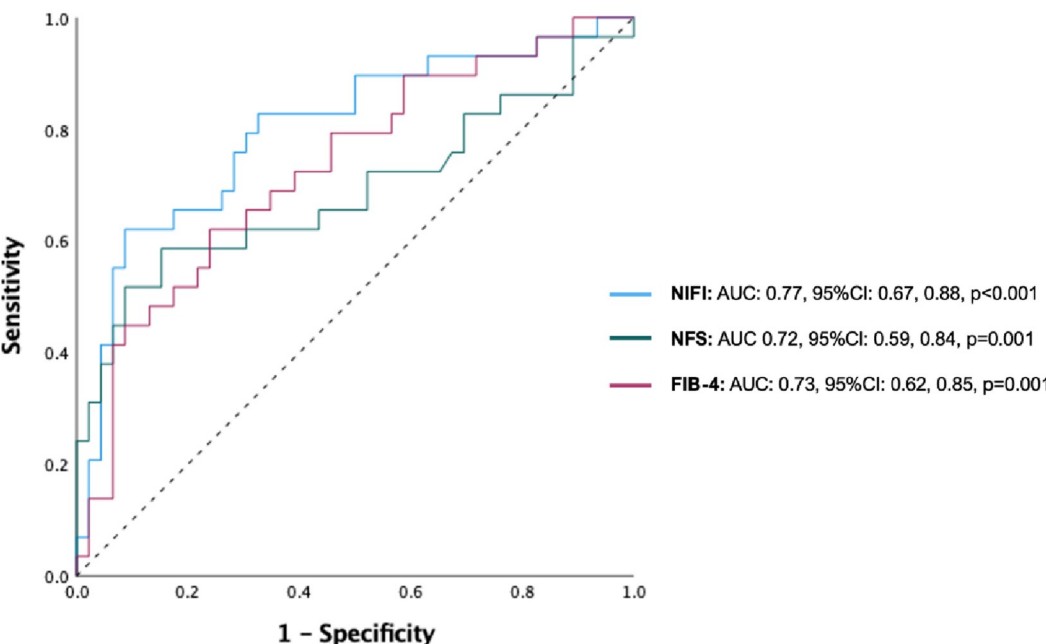

**Fig 1. Receiver-operating characteristic (ROC) curves showing area under the curve (AUC) for 105 patients.** The outcome of NAFLD severity (defined by Fibroscan score ≥7.2 kPa) is tested against NAFLD fibrosis index (NIFI), NAFLD fibrosis score (NFS) and FIB-4 score.

number of metabolic markers in NAFLD patients with more significant disease when compared with those with less severe disease based on fibroscan findings.

IL-6 is a pro-inflammatory cytokine adipocyte derived pro-inflammatory cytokine produced by adipocytes and hepatocytes. IL-6 impairs insulin signalling in hepatocytes, increases gluconeogenesis resulting in compensatory hyperinsulinaemia [4]. IL-6 is thought to increase insulin resistance through its inhibition of cytokine signalling [36]. In the present study we found increased IL-6 concentrations in patients with significant fibrosis as evidenced by fibroscan scores which supports the results of previous studies that found higher IL-6 concentrations in NAFLD patients with advanced histological disease [32] as well as showing that sustained exposure to IL-6 sensitises the liver to injury and promotes apoptosis [37].

IL-6 concentrations are raised in a number of systemic inflammatory conditions including rheumatoid arthritis (RA). However, the magnitude of elevation in IL-6 concentrations is many fold higher in these inflammatory conditions than in our NAFLD population [38]. This could reflect that our population group were relatively well and the fact that we had excluded patients with any systemic inflammatory conditions. Despite a relatively healthy patient cohort, we detected a significant difference in IL-6 concentrations between patients with varying NAFLD severity in our study.

Elevated IL-6 concentrations are also associated with CV risk. IL-6 has a number of properties that foster development of CV disease, including endothelial cell activation and smooth muscle proliferation [39]. It is possible that our findings simply reflect the increasing CV risk patients with more advanced fibrosis, rather than IL-6 playing a causative role in NAFLD pathogenesis. We did find an association in the multivariable analysis between age and IL-6 concentrations (S3b Table) which could offer another explanation for our findings. Further studies will be needed to elucidate our findings. We did not detect an association between IL-6 concentrations and patients with established CV disease in our NAFLD population, however this was because only 9.5% (Table 1) of our patient group had established CV disease.

A number of novel pharmacotherapy agents against NAFLD are in the pipeline such as obeticholic acid (farnesoid X receptor agonist) [40]. IL-6 blockade may offer another potential therapeutic channel for NAFLD treatment. In fact, blockade of IL-6 is the mechanism of action of drugs such as tocilizumab, which is a recombinant humanised anti-human IL-6 receptor monoclonal antibody used in the treatment of rheumatoid arthritis [38].

MMP-9 is a member of a family of zinc-dependent neutral proteases that degrade the extracellular matrix and basement membrane and has been implicated in sinusoidal injury in liver failure, liver remodelling and necrosis [23]. Previously, there have been no direct studies in NAFLD patients assessing the role of MMP-9 which is involved in liver injury and remodeling [23]. The present study shows a decrease in MMP-9 with more advanced fibrosis, these findings are consistent with a previous study that revealed lower concentrations of MMP-9 in hepatitis C patients with advanced fibrosis [41]. A similar pathophysiological involvement of this protease in the fibrogenesis underlying the progression of NAFLD may be an explanation.

IL-6 and MMP-9 results in this study may reflect the two important pathological processes underlying NAFLD, inflammation and fibrosis, respectively. Thus, combining the 2 markers to generate the NIFI may give superior predictive outcomes for disease severity than a single marker alone. There is a need to prospectively evaluate this index. Whether this index is superior to the currently employed FIB-4 and ELF score needs to be determined. Where such an index would fit into the diagnostic and management pathway for NAFLD would also need to be assessed. The NIFI marker may potentially enable assessment of patients with NAFLD at more frequent intervals than is possible with serial liver biopsies due to the invasive nature of the procedure as well as the logistical and financial considerations involved in obtaining and interpreting a liver biopsy. It may also overcome the sampling issues associated with liver

biopsy where there is a potential of misclassification due to the fact that fibrosis is not evenly distributed throughout the liver.

Ghrelin and TNFα concentrations in this study were positively associated with more advance fibrosis as evidenced by Fibroscan values. These findings are in agreement with previous studies but were not significant after multivariate analysis. In fact, by multivariate analysis, only IL-6 and MMP9 were independently associated with fibrosis. An explanation for this may be the population size in the present study.

This study did not find significantly different concentrations of adiponectin, leptin, resistin or PAI-1 in patients with varying stages of NAFLD as evidenced by fibroscan scores. These metabolic markers have previously been shown to play a role in NAFLD, but our study findings are not unique as other researchers have also failed to find an association [21,26–28]. Again, this may be a due to the size of the present study.

In the present study, no significant difference was found between keratin-18 concentration and increased fibroscan values. This is contrary to previous studies [25,42,43] and a recent systematic review [44] exploring the role of keratin-18 for fibrosis in NAFLD. The reason for this is not clear but could be due to a type 2 error and our sample size or how the data was categorised in the current study. Keratin-18 serum levels were indeed higher in the patients with fibroscan value ≥7.2 kPa but did not reach statistical significance. Comparing the continuous serum-based markers data with categorised Fibroscan results rather than continuous fibroscan scores could be an explanation for not detecting an association with keratin-18 and indeed some of the other markers that did not reach statistical significance.

This study did not find significantly different concentrations of sIL-6R, sTNFR1 or sTNFR2 with fibroscan ≥7.2 kPa. Although a previous study found increased concentrations the sIL-6R and sTNFR1 in NAFLD patients compared with healthy volunteers [45], this is the first study investigating these 3 receptors in different stages of NAFLD. The correlation found in the current study with these receptors and liver aminotransferases (Table 2) is consistent with the findings the pervious study mentioned above [45].

## Study limitations

The limitations of the current study included a lack of patients who had a liver biopsy; hence a lack of a gold standard to which the NIFI can be compared. We also had relatively few patients with established cardiovascular disease which is the leading cause for mortality in the NAFLD population, and this may be the reason why we did not find an association between the serum-based markers and history of cardiovascular disease. The current study serum-based markers were not measured and compared to matched healthy population.

## Conclusions

IL-6 and MMP-9 are differentially expressed with increasing severity of NAFLD. Although further validation studies are needed, the NIFI, in combination with Fibroscan, could potentially serve as a marker for NAFLD severity and could prove to be a more acceptable and cost-effective alternative to liver biopsy. Given the role played in cardiovascular disease by the constituent markers of the NIFI, it could also potentially, serve as a predictor of vascular risk.

## Supporting information

**S1 Table. Geometric mean (95% CI) levels of serum-based markers, according to demographic factors.**
(DOCX)

**S2 Table.** a. Geometric Mean (95% CI) levels of serum-based markers, according to Fibroscan result. p-values calculated using unpaired t-test (of log values)–no formal correction made for multiple testing. b. Geometric Mean (95% CI) levels of serum-based markers, according to CAP result. p-values calculated using unpaired t-test (of log values)–no formal correction made for multiple testing.
(DOCX)

**S3 Table.** a. Demographic factors associated with ghrelin concentrations in multivariable model. Results from linear regression model (log scale). Fit of model determined by examination of Pearson residual plots. b. Demographic factors associated with interleukin-6 (IL-6) concentrations in multivariable model. Results from linear regression model (log scale). Fit of model determined by examination of Pearson residual plots. c. Demographic factors associated with tumor necrosis factor a (TNFa) concentrations in multivariable model. Results from linear regression model (log scale). Fit of model determined by examination of Pearson residual plots. d. Demographic factors associated with metalloproteinase 9 (MMP9) concentrations in multivariable model. Results from linear regression model (log scale). Fit of model determined by examination of Pearson residual plots.
(DOCX)

**S4 Table.** a. Factors associated with Fibroscan result ≥9.6 kPa. Logistic regression analysis in all study patients. b. Multi-variable analysis of factors associated with Fibroscan result ≥9.6 kPa in all study patients.
(DOCX)

**S1 File.**
(DOCX)

**S2 File. Dataset table.**
(XLSX)

## Author Contributions

**Conceptualization:** Marta Guerrero Misas, Devaki Nair, Emmanuel Tsochatzis.

**Data curation:** Atul Goyale, Anjly Jain, Davide Roccarina, Laura Iogna Prat.

**Formal analysis:** Atul Goyale, Anjly Jain, Colette Smith.

**Supervision:** Devaki Nair, Emmanuel Tsochatzis.

**Writing – original draft:** Atul Goyale.

**Writing – review & editing:** Atul Goyale, Margarita Papatheodoridi, Dimitri P. Mikhailidis, Devaki Nair, Emmanuel Tsochatzis.

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
