## [Decision Letter · Decision Letter 0]

18 Aug 2021

PONE-D-21-17417

Assessment of Non-alcoholic Fatty Liver Disease (NAFLD) severity with novel metabolic markers: A Pilot Study

PLOS ONE

Dear Dr. Papatheodoridi,

Thank you for submitting your manuscript to PLOS ONE. After careful consideration, we feel that it has merit but does not fully meet PLOS ONE’s publication criteria as it currently stands. Therefore, we invite you to submit a revised version of the manuscript that addresses the points raised during the review process.

While both reviewers evaluated your work as potentially interesting, they also described several important issues that need to be addressed. 

We look forward to receiving your revised manuscript.

Kind regards,

Pavel Strnad

Academic Editor

PLOS ONE

Journal Requirements:

2.  Thank you for submitting the above manuscript to PLOS ONE. During our internal evaluation of the manuscript, we found significant text overlap between your submission and the following previously published works.

- https://linkinghub.elsevier.com/retrieve/pii/S002191501830491X

- https://journals.sagepub.com/doi/full/10.1177/0004563214523739

We would like to make you aware that copying extracts from previous publications, especially outside the methods section, word-for-word is unacceptable, even for works which you authored. In addition, the reproduction of text from published reports has implications for the copyright that may apply to the publications.

Please revise the manuscript to rephrase the duplicated text, cite your sources, and provide details as to how the current manuscript advances on previous work. Please note that further consideration is dependent on the submission of a manuscript that addresses these concerns about the overlap in text with published work.

5. Please include a copy of Table 4A which you refer to in your text on page 17.

Reviewers' comments:

Reviewer's Responses to Questions

**Comments to the Author**

1. Is the manuscript technically sound, and do the data support the conclusions?

Reviewer #1: Yes

Reviewer #2: Partly

2. Has the statistical analysis been performed appropriately and rigorously? 

Reviewer #1: Yes

Reviewer #2: Yes

3. Have the authors made all data underlying the findings in their manuscript fully available?

Reviewer #1: Yes

Reviewer #2: Yes

4. Is the manuscript presented in an intelligible fashion and written in standard English?

Reviewer #1: Yes

Reviewer #2: Yes

5. Review Comments to the Author

Reviewer #1: Goyale et al. analysed 12 different adipokines and cytokines as biomarker in a cohort of 105 clinically diagnosed NAFLD patients. The manuscript is well written. The method section includes a detailed description of the statistical analysis including a power analysis for this study. IL-6 significantly increased and MMP9 significantly decreased with increasing fibrosis severity as determined by Fibroscan. A newly generated NAFLD individual fibrosis index (NIFI) showed a good sensitivity, but a mediocre specificity for fibrosis severity. No validation data are available for NIFI. Results of IL-6 and MMP9 are of interest for better understanding of disease progression.

Please find my comments below:

Major comments:

- in Table 1 NFS is depicted, but no data for FIB-4. How many patients had a FIB-4 >1.3?

- Are there also data for CAP measurements available in this cohort?

- Univariate and multivariate analysis were performed for factors associated with Fibroscan > 7.2kPa. Was univariate and multivariate analysis also performed for further Fibroscan levels, e.g. Fibroscan> 9.6kPa for advanced disease?

- In this study no significant associations between adiponectin, leptin, resistin or PAI-1 and liver fibrosis was found. Were these metabolic markers also analysed with “metabolic” disease feature, e.g. CAP values?

- Figure 1: Only ROC of NIFI is depicted. Further ROCs for FIB-4 and NFS in the analysed cohort should be added.

- Higher IL-6 leves were significantly associated to Fibroscan > 7.2kPa. Are there any data for other inflammatory markers available? e.g. hsCRP as another inflammatory marker that is associated with CVD events?

- Follow-up data for this patient cohort would be interesting to see the development of NIFI over time.

- Further validation cohort for NIFI were desirable.

Minor comments:

- In the Method section inclusion and exclusion criteria were described, but in line 115 only patients on TNF a inhibitor therapy were mentioned as excluded. How about other patients with immunosuppressive treatment? Were they also excluded?

- line 366: Elafibranor has failed the phase 3 study. Please update this sentence

- line317: calculated…

Reviewer #2: In the present study the authors investigated 12 blood markers in patients with non-biopsy-proven NAFLD (n=105). They found that IL-6 and MMP-9 blood levels were associated with higher VCTE values (≥ 7.2 kPa). Based on this finding they established an index (NIFI) which predicted VCTE ≥ 7.2 kPa with moderate AUC (0.77).

To improve the clinical relevance of the study, the following points should be addressed:

1) The marker panel represents not only metabolic parameters but also markers of inflammation, cell death and fibrosis. The term “metabolic markers” is therefore not correct and should be replaced.

2) The authors should provide information whether they used serum or plasma for the analyses of the different markers.

3) The term “geometric mean” should be explained.

4) It remains unclear which keratin-18 assay (M30 or M65 ELISA) was used. The authors should also provide information about the role of K18 biomarkers in NAFLD and explain why, e.g., the M30 was preferred over the M65 biomarker. The term cytokeratin-18 is no longer used and should be replaced with keratin-18.

5) For VCTE, cut-off values of 8 kPa and 12 kPa are recommended for risk stratification, i.e. rule out or rule in advanced fibrosis, in NAFLD (EASL Clinical Practice Guidelines on non-invasive tests for evaluation of liver disease severity and prognosis – 2021 update; J Hepatol 2021). The authors should therefore use these cut-off values for their analyses. It would be interesting to analyze how the markers correlate with advanced NAFLD. In this respect the different markers should be compared with VCTE values above and below 8 kPa or 12 kPa.

6) Since FIB-4 or NFS are recommended for risk stratification of NAFLD in primary care, the evaluated markers should also be compared with FIB-4 values above and below 1.3 or 2.67 and NFS values above and below -1.455 or 0.675.

7) If histological data obtained from a liver biopsy closely performed to the measurement of the blood markers are available, a sub-analysis comparing the different markers with the diagnosis NASH versus NAFL as well as with the NAFLD activity and the single NAS components and fibrosis stages would be interesting.

8) Aminotransferase levels are not suitable for the detection of disease activity in NAFLD, e.g. a significant proportion of NASH patients reveal normal aminotransferase levels despite progressed NAFLD. A correlation of the markers with aminotransferases is therefore less informative which should be considered and discussed.

9) The authors stated that VCTE ≥7.2 kPa was significantly associated with increased IL-6 concentrations and lower MMP-9 concentrations. However, in table 3b, it is indicated that similarly to IL-6, MMP-9 levels per 1-log “higher” were significantly associated with VCTE ≥ 7.2 kPa, which should be clarified.

10) Since more than 50% of patients reveal a BMI of > 30, more information about the success rate and quality criteria are required for VCTE measurements with the M probe in this patient cohort. It should be mentioned whether XL-probe was available or not.

11) More information about the calculation of the NIFI index is required. HOMA-IR calculation was firstly described 1985 by Matthews D.R. et al. in Diabetologia. The authors should refer to this publication with respect to the citation of HOMA-IR calculation.

In addition to sensitivity and specificity, information about the PPV and NPV should be provided. The specificity of this index is rather low and the AUC remains below 0.8. Moreover, ~25% of patients remain between the 2 selected cut-offs. To further evaluate the diagnostic performance of the NIFI index, it should be compared to FIB-4 and NFS. Information about the number of NAFLD patients with intermediate results for FIB-4 or NFS should be provided.

12) In the abstract (results), information about the outcome prediction by ROC, i.e. VCTE ≥ 7.2 kPa, is missing.

13) The markers evaluated in this study should be simultaneously compared in matched healthy individuals.

14) Further corrections required: page 19, lane 317: NIFI was calculated by…. Page 22, lane 394, page 23, lane 404 and 408: fibroscan values (instead of scores). Reference 25: Feldstein et al. Page 24: The current study metabolic markers were not measured (in the current study…). Some reference numbers are in bold.

6. PLOS authors have the option to publish the peer review history of their article (what does this mean?). If published, this will include your full peer review and any attached files.

Reviewer #1: No

Reviewer #2: No

---

## [Author Response · Author response to Decision Letter 0]

11 Oct 2021

Manuscript Ref. PONE-D-21-17417

Dear Dr Strnad,

We would like to thank you for giving us the opportunity to submit a revised version of our manuscript entitled “Assessment of non-alcoholic fatty liver disease (NAFLD) severity with novel serum based markers: A Pilot Study” for consideration for publication in Plos One journal. 

We appreciate the time and effort that you and the reviewers dedicated to providing feedback on our manuscript and are grateful for the insightful comments and valuable improvements to our paper. We believe that after completion of the suggested edits, the revised manuscript has benefitted from an improvement in overall presentation and clarity. To this end, we have incorporated the suggestions made by the reviewers to the best extent possible. Those changes are underlined within the main manuscript as well as supplementary material. 

In addition, we have now provided our dataset as a separate supplementary file, as requested and we would like to have our data availability status updated.

In our revised main manuscript and supplementary material, you will find all the changes/new additions in red font.

Below you will find, in blue, a point-by-point response to the reviewers’ comments and concerns. All page numbers refer to the revised manuscript files.

Journal Requirements

We would like to thank you for your comment. We have now updated our manuscript format according to the journal’s requirements. 

2. Thank you for submitting the above manuscript to PLOS ONE. During our internal evaluation of the manuscript, we found significant text overlap between your submission and the following previously published works.

- https://linkinghub.elsevier.com/retrieve/pii/S002191501830491X

- https://journals.sagepub.com/doi/full/10.1177/0004563214523739

We would like to make you aware that copying extracts from previous publications, especially outside the methods section, word-for-word is unacceptable, even for works which you authored. In addition, the reproduction of text from published reports has implications for the copyright that may apply to the publications.

Please revise the manuscript to rephrase the duplicated text, cite your sources, and provide details as to how the current manuscript advances on previous work. Please note that further consideration is dependent on the submission of a manuscript that addresses these concerns about the overlap in text with published work.

Thank you for this comment. The first linked publication refers to an abstract presented by us in an international congress showing the results of the work that is fully described in this manuscript. Therefore, we do not consider that this may constitute plagiarism. If you still consider we should rephrase our abstract, we can happily make the necessary changes. 

With regards to the second linked file, this is on an entirely different topic. We read through the manuscript but we could not detect any similarities. We would be grateful if you could point these out. 

Thank you for the opportunity to share our data. We have now uploaded our dataset as Supporting information and we would like to have our data availability status updated. 

4. Please note that in order to use the direct billing option the corresponding author must be affiliated with the chosen institute. Please either amend your manuscript to change the affiliation or corresponding author or email us at plosone@plos.org with a request to remove this option.

The corresponding author, Prof. E. Tsochatzis, is affiliated to UCL. Please let us know if you require further information. 

5. Please include a copy of Table 4A which you refer to in your text on page 17.

Thank you for pinpointing this typo. The table referred to at this point is Table 3a and there is no Table 4a for this manuscript. We have now changed this in our main manuscript, page 15. 

Thank you for this comment. We have now amended the issues with in-text citations and reference style, while you may find the captions for the Supporting information at the end of our main manuscript. 

Reviewers’ Comments to the Author

Reviewer #1: Goyale et al. analysed 12 different adipokines and cytokines as biomarker in a cohort of 105 clinically diagnosed NAFLD patients. The manuscript is well written. The method section includes a detailed description of the statistical analysis including a power analysis for this study. IL-6 significantly increased and MMP9 significantly decreased with increasing fibrosis severity as determined by Fibroscan. A newly generated NAFLD individual fibrosis index (NIFI) showed a good sensitivity, but a mediocre specificity for fibrosis severity. No validation data are available for NIFI. Results of IL-6 and MMP9 are of interest for better understanding of disease progression.

Please find my comments below:

Major comments:

- in Table 1 NFS is depicted, but no data for FIB-4. How many patients had a FIB-4 >1.3?

Thank you for your comment. Number of patients (%) who had FIB-4 > 1.3 has been now added in Table 1 (page 11-12). 

- Are there also data for CAP measurements available in this cohort?

Thank for raising this point. Yes, there are data for CAP measurements in this cohort. Number of patients (%) who had CAP >302 dB/m has been now added in Table 1 (page 12), as this cut-off has been considered to predict S1 steatosis and it is explained in Methods section (ref 31) (page 8).

- Univariate and multivariate analysis were performed for factors associated with Fibroscan > 7.2kPa. Was univariate and multivariate analysis also performed for further Fibroscan levels, e.g. Fibroscan> 9.6kPa for advanced disease? 

Thank for this comment. We have now performed univariate and multivariate analysis for factors associated with Fibroscan> 9.6kPa for advanced fibrosis and this is now presented in S4a and S4b Tables. Moreover, we discussed the findings of this analysis on page 16 of our main manuscript. 

- In this study no significant associations between adiponectin, leptin, resistin or PAI-1 and liver fibrosis was found. Were these metabolic markers also analysed with “metabolic” disease feature, e.g. CAP values?

We would like to thank for this comment. We have now analysed all the metabolic and other serum markers in relation to CAP values and this analysis can be now found in Table S2b and page 14 of the main manuscript.

- Figure 1: Only ROC of NIFI is depicted. Further ROCs for FIB-4 and NFS in the analysed cohort should be added.

Thank you for this comment. This is now added in the updated Figure 1, pages 17-18 of the main manuscript.

- Higher IL-6 levels were significantly associated to Fibroscan > 7.2kPa. Are there any data for other inflammatory markers available? e.g. hsCRP as another inflammatory marker that is associated with CVD events? 

We understand that examining more inflammatory markers would be optimal, however we do not have enough available data regarding other inflammatory markers, such as hsCRP, in our dataset.

- Follow-up data for this patient cohort would be interesting to see the development of NIFI over time. 

We agree with the reviewer that follow-up data would be needed to assess the development of NIFI over time. This is this is a prospective cohort, however no clinical events have been developed to date. We have not performed repeat measurements of the biomarkers to date but we will certainly do in the future.

- Further validation cohort for NIFI were desirable.

We agree with the reviewer at this point and we have already mentioned this issue as a limitation in our discussion section.

Minor comments:

- In the Method section inclusion and exclusion criteria were described, but in line 115 only patients on TNFa inhibitor therapy were mentioned as excluded. How about other patients with immunosuppressive treatment? Were they also excluded? 

Thank you for giving us the opportunity to clarify this. All patients on immunosuppressive therapy were excluded. This is now updated on page 7 of the main manuscript. 

- line 366: Elafibranor has failed the phase 3 study. Please update this sentence. 

Thank you for pinpointing this out. This has been now removed from the sentence on page 20 of the main manuscript.

- line317: calculated…

Thank you for pinpointing this mistake. We have now corrected the typo on page 27. 

Reviewer #2: In the present study the authors investigated 12 blood markers in patients with non-biopsy-proven NAFLD (n=105). They found that IL-6 and MMP-9 blood levels were associated with higher VCTE values (≥ 7.2 kPa). Based on this finding they established an index (NIFI) which predicted VCTE ≥ 7.2 kPa with moderate AUC (0.77).

To improve the clinical relevance of the study, the following points should be addressed:

1) The marker panel represents not only metabolic parameters but also markers of inflammation, cell death and fibrosis. The term “metabolic markers” is therefore not correct and should be replaced. 

Thank you for this comment. We have now updated the term “metabolic markers” with serum-based markers, which can better reflect the wide spectrum of the different markers assessed in our study. This has been changed throughout the main manuscript and supporting information.

2) The authors should provide information whether they used serum or plasma for the analyses of the different markers. 

Serum samples were used for the analysis of these markers and this is now clarified in page 9 of the main manuscript. 

3) The term “geometric mean” should be explained.

Thank you for this comment. The term “geometric mean” is now described on page 10 of the main manuscript. 

4) It remains unclear which keratin-18 assay (M30 or M65 ELISA) was used. The authors should also provide information about the role of K18 biomarkers in NAFLD and explain why, e.g., the M30 was preferred over the M65 biomarker. The term cytokeratin-18 is no longer used and should be replaced with keratin-18.

Thank you for this comment. We used the M30 assay and this is now clarified in the methods section in page 11. The term cytokeratin-18 has been replaced by the term keratin-18 as requested.

The M30 antibody identifies a fragmented form of keratin 18, which is an apoptosis-specific neo-epitope at the keratin aspartic acid residue 396, generated by caspase-6, caspase-3 and caspase-7 cleavage. The M65 antibody allows for measurement of all keratin 18 fragments because of loss of cell membrane integrity from necrosis and/or apoptosis. The relevant mechanism of injury in NAFLD is apoptosis, therefore we used the M30 assay also in line with several other studies on the topic.

5) For VCTE, cut-off values of 8 kPa and 12 kPa are recommended for risk stratification, i.e. rule out or rule in advanced fibrosis, in NAFLD (EASL Clinical Practice Guidelines on non-invasive tests for evaluation of liver disease severity and prognosis – 2021 update; J Hepatol 2021). The authors should therefore use these cut-off values for their analyses. It would be interesting to analyze how the markers correlate with advanced NAFLD. In this respect the different markers should be compared with VCTE values above and below 8 kPa or 12 kPa.

Thank you for this comment. We were looking for significant rather than advanced fibrosis and therefore used the 7.2 KPa cut-off. We have now analysed the 9.6 cut-off for advanced fibrosis as requested by reviewer 1, and this is included in the S4a and S4b Tables while it is discussed on page 20 of the main manuscript. We used a single TE cut-off (with maximum combined se and sp) based in published literature rather than the rule-in and rule-out cut-offs that we recently described in a publication. 

6) Since FIB-4 or NFS are recommended for risk stratification of NAFLD in primary care, the evaluated markers should also be compared with FIB-4 values above and below 1.3 or 2.67 and NFS values above and below -1.455 or 0.675.

We would like to thank you for this comment. This is a secondary care cohort and most patients already had FIB4 >1.3 before referral from primary care. Moreover, our newly developed NIFI index aims to predict significant rather than advanced fibrosis. Taking into account these caveats, we did perform an analysis which shows that the sensitivity levels to rule-out significant fibrosis are much lower (66%) than NIFI for both FIB-4 and NFS, while the rule-in specificity reaches 100%. This sounds reasonable, as these cut-offs were initially optimised for predicting advanced but not significant fibrosis. This is now discussed in Results section, page 18.

7) If histological data obtained from a liver biopsy closely performed to the measurement of the blood markers are available, a sub-analysis comparing the different markers with the diagnosis NASH versus NAFL as well as with the NAFLD activity and the single NAS components and fibrosis stages would be interesting.

We agree with the reviewer that this would be an interesting sub-analysis. However, unfortunately we do not have enough histological data for such analysis.

8) Aminotransferase levels are not suitable for the detection of disease activity in NAFLD, e.g. a significant proportion of NASH patients reveal normal aminotransferase levels despite progressed NAFLD. A correlation of the markers with aminotransferases is therefore less informative which should be considered and discussed.

We certainly agree with this comment – it is also possible that the normal levels in NAFLD are lower than what is commonly accepted. However, we use the transaminases as continuous variables and we do think that these correlations are of relative value. We briefly discuss this on page 12 of the main manuscript. 

9) The authors stated that VCTE ≥7.2 kPa was significantly associated with increased IL-6 concentrations and lower MMP-9 concentrations. However, in table 3b, it is indicated that similarly to IL-6, MMP-9 levels per 1-log “higher” were significantly associated with VCTE ≥ 7.2 kPa, which should be clarified.

As presented in table 3b, higher levels of log10 IL-6 were significantly and positively associated with VCTE ≥ 7.2 kPa (OR:2.34, p=0.008), while higher levels of log10 MMP-9 were significantly but inversely associated with VCTE ≥7.2 kPa (OR 0.34, p=0.008).

10) Since more than 50% of patients reveal a BMI of > 30, more information about the success rate and quality criteria are required for VCTE measurements with the M probe in this patient cohort. It should be mentioned whether XL-probe was available or not. 

XL probe was available and performed as recommended by the device. This is now clarified in methods section, page 8.

11) More information about the calculation of the NIFI index is required. HOMA-IR calculation was firstly described 1985 by Matthews D.R. et al. in Diabetologia. The authors should refer to this publication with respect to the citation of HOMA-IR calculation.

In addition to sensitivity and specificity, information about the PPV and NPV should be provided. The specificity of this index is rather low and the AUC remains below 0.8. Moreover, ~25% of patients remain between the 2 selected cut-offs. To further evaluate the diagnostic performance of the NIFI index, it should be compared to FIB-4 and NFS. Information about the number of NAFLD patients with intermediate results for FIB-4 or NFS should be provided.

Thank you for this comment. This is now presented in the updated Figure 1 (page 17-18), which includes the ROC curves for FIB-4 and NFS, as requested by the reviewer 1 as well. As it is shown, FIB-4 and NFS have a lower AUC than NIFI. Number of patients with intermediate results for both FIB-4 and NFS are now provided in results section page 17.

12) In the abstract (results), information about the outcome prediction by ROC, i.e. VCTE ≥ 7.2 kPa, is missing.

Thank you for this comment. We have now added the relevant findings (page 2). 

13) The markers evaluated in this study should be simultaneously compared in matched healthy individuals.

We agree that it would be interesting to compare the levels of these markers between NAFLD and healthy individuals. However, we have not included any healthy individuals in this cohort.

14) Further corrections required: page 19, lane 317: NIFI was calculated by…. Page 22, lane 394, page 23, lane 404 and 408: fibroscan values (instead of scores). Reference 25: Feldstein et al. Page 24: The current study metabolic markers were not measured (in the current study…). Some reference numbers are in bold.

Thank you for pinpointing these mistakes. We have now made all necessary corrections in our revised manuscript.

---

## [Decision Letter · Decision Letter 1]

3 Nov 2021

PONE-D-21-17417R1Assessment of Non-alcoholic Fatty Liver Disease (NAFLD) severity with novel serum-based markers: A Pilot StudyPLOS ONE

Dear Dr. Papatheodoridi,

Thank you for submitting your manuscript to PLOS ONE. After careful consideration, we feel that it has merit but does not fully meet PLOS ONE’s publication criteria as it currently stands. Therefore, we invite you to submit a revised version of the manuscript that addresses the points raised during the review process.

As you can see, both reviewers were satisfied with the improvements that you made and only minor changes are required at this stage.

We look forward to receiving your revised manuscript.

Kind regards,

Pavel Strnad

Academic Editor

PLOS ONE

Journal Requirements:

Reviewers' comments:

Reviewer's Responses to Questions

**Comments to the Author**

1. If the authors have adequately addressed your comments raised in a previous round of review and you feel that this manuscript is now acceptable for publication, you may indicate that here to bypass the “Comments to the Author” section, enter your conflict of interest statement in the “Confidential to Editor” section, and submit your "Accept" recommendation.

Reviewer #1: (No Response)

Reviewer #2: (No Response)

2. Is the manuscript technically sound, and do the data support the conclusions?

Reviewer #1: Yes

Reviewer #2: Yes

3. Has the statistical analysis been performed appropriately and rigorously? 

Reviewer #1: Yes

Reviewer #2: Yes

4. Have the authors made all data underlying the findings in their manuscript fully available?

Reviewer #1: Yes

Reviewer #2: Yes

5. Is the manuscript presented in an intelligible fashion and written in standard English?

Reviewer #1: Yes

Reviewer #2: Yes

6. Review Comments to the Author

Reviewer #1: The authors addressed the majority of the raised comments in their revision and performed the suggested additional analyses. The manuscript has improved by this revision.

Reviewer #2: The comments of the reviewers have been mainly addressed; however, some minor corrections are still required:

1. Discussion page 22: "a recent systemic review...." replace reference 41 with reference 42.

2. Discussion page 22: "This is contrary to previous studies and a recent systemic review exploring the role keratin-18 in NAFLD". This sentence should be corrected since the mentioned study (not studies) or review article did not evaluate or discuss an association of keratin-18 with fibroscan values but of keratin-18 fragments with histological fibrosis. It would be better to write “the role of keratin-18 for fibrosis in NAFLD”. Further studies indicating a correlation of keratin-18 fragments with fibrosis progression in NAFLD might be considered (Tamimi T et al., J Hepatol 2011; 54:1224-29; Diab DL et al., Clinical Gastroenterol Hepatol 2008; 6(11):1249-54).

3. S1 and S2aTables: Replace Cytokeratin-18 with Keratin-18.

7. PLOS authors have the option to publish the peer review history of their article (what does this mean?). If published, this will include your full peer review and any attached files.

Reviewer #1: No

Reviewer #2: No

---

## [Author Response · Author response to Decision Letter 1]

4 Nov 2021

RESPONSE TO REVIEWERS

Manuscript Ref. PONE-D-21-17417

Dear Dr Strnad,

We would like to thank you for your consideration of our manuscript entitled “Assessment of non-alcoholic fatty liver disease (NAFLD) severity with novel serum based markers: A Pilot Study” with minor revisions. 

We have tried to address all comments from the reviewers and we are looking forward to your final decision regarding the publication of our study. 

Below you will find, in blue, a point-by-point response to the reviewers’ comments and concerns. All page numbers refer to the revised manuscript files.

In our revised main manuscript and supplementary material, you will find all the changes/new additions in red font.

Journal Requirements

We have reviewed our reference list carefully and we have not found any retracted article.

We have now revised the order of the citations, following the addition of 2 more references, according to Reviewer’s comment no 2, as you will see in the revised manuscript. 

Reviewers’ Comments to the Author

Reviewer #1: The authors addressed the majority of the raised comments in their revision and performed the suggested additional analyses. The manuscript has improved by this revision.

Reviewer #2: The comments of the reviewers have been mainly addressed; however, some minor corrections are still required:

1. Discussion page 22: "a recent systemic review...." replace reference 41 with reference 42.

Thank you for your comment. We have now corrected this to cite the correct study, which is no 44 in the updated reference list, after the additions mentioned in the next comment. 

2. Discussion page 22: "This is contrary to previous studies and a recent systemic review exploring the role keratin-18 in NAFLD". This sentence should be corrected since the mentioned study (not studies) or review article did not evaluate or discuss an association of keratin-18 with fibroscan values but of keratin-18 fragments with histological fibrosis. It would be better to write “the role of keratin-18 for fibrosis in NAFLD”. Further studies indicating a correlation of keratin-18 fragments with fibrosis progression in NAFLD might be considered (Tamimi T et al., J Hepatol 2011; 54:1224-29; Diab DL et al., Clinical Gastroenterol Hepatol 2008; 6(11):1249-54).

Thank you for this comment. We have now updated this part of our discussion and we have added the suggested references as requested, (ref no 42, 43).

3. S1 and S2aTables: Replace Cytokeratin-18 with Keratin-18.

Thank you for this comment. We have now amended the typo in the updated Suppl. Material.

---

## [Editor Report · Decision Letter 2]

8 Nov 2021

Assessment of Non-alcoholic Fatty Liver Disease (NAFLD) severity with novel serum-based markers: A Pilot Study

PONE-D-21-17417R2

Dear Dr. Papatheodoridi,

We’re pleased to inform you that your manuscript has been judged scientifically suitable for publication and will be formally accepted for publication once it meets all outstanding technical requirements.

Kind regards,

Pavel Strnad

Academic Editor

PLOS ONE
---

## [Editor Report · Acceptance letter]

15 Nov 2021

PONE-D-21-17417R2 

Assessment of Non-alcoholic Fatty Liver Disease (NAFLD) severity with novel serum-based markers: A Pilot Study 

Dear Dr. Papatheodoridi:

I'm pleased to inform you that your manuscript has been deemed suitable for publication in PLOS ONE. Congratulations! Your manuscript is now with our production department. 

Kind regards, 

on behalf of

Dr. Pavel Strnad 

Academic Editor

PLOS ONE